# Knowledge, understanding and perceptions of key stakeholders on the maternity protection available and accessible to female domestic workers in South Africa

Catherine Pereira-Kotze[1]*, Mieke Faber[2,3], Tanya Doherty[1,4]

**1** School of Public Health, Faculty of Community and Health Sciences, University of the Western Cape, Cape Town, South Africa, **2** Non-Communicable Diseases Research Unit, South African Medical Research Council, Cape Town, South Africa, **3** Department of Dietetics and Nutrition, Faculty of Community and Health Sciences, University of the Western Cape, Cape Town, South Africa, **4** Health Systems Research Unit, South African Medical Research Council, Cape Town, South Africa

* catherinejanepereira@gmail.com

## Abstract

Maternity protection enables women to combine reproductive and productive roles. Domestic workers are a vulnerable group due to heterogeneous non-standard employment relationships and are unlikely to have access to comprehensive maternity protection. This study aimed to explore the knowledge, understanding and perceptions of key stakeholders in government, trade unions, non-governmental organisations and other relevant organisations of the maternity protection entitlements that should be available and accessible to female domestic workers in South Africa. This qualitative cross-sectional study included in-depth interviews with fifteen stakeholders working in different sectors in South Africa and mainly at a national level involved in maternity protection availability and access. Results show that stakeholders appear to have limited understanding of comprehensive maternity protection. Many challenges related to accessing cash payments while on maternity leave were described and suggestions were provided for how this could be improved. Participants described how certain labour-related characteristics unique to the domestic work sector were barriers in accessing maternity protection. Ensuring greater awareness of all components of maternity protection and improving implementation of existing labour legislation intended to guarantee maternity protection for non-standard workers in South Africa is important to improve access to maternity protection for this vulnerable group. Improved access to maternity protection would contribute to optimal maternal and new-born health and ensure economic security for women around the time of childbirth.

## Introduction

The provision of maternity protection at work enables women to combine their reproductive and productive roles and improves gender equality in the workplace [1]. The International Labour Organization (ILO) defines the main elements of comprehensive maternity protection

**Data Availability Statement:** All data relevant to the paper has been presented within the paper

itself (in the form of participant quotations). The datasets generated and/or analysed during the current study are not publicly available to protect participant confidentiality but are available in an anonymised form from the corresponding author on reasonable request. In the interest of maintaining long-term data accessibility, the University of the Western Cape Research Office can also be contacted for data requests at: research-office@uwc.ac.za.

**Funding:** CPK received support from the DSI/NRF Centre of Excellence in Food Security UID 91490. TD was supported by the South African Medical Research Council. The funders had no role in study design, data collection and analysis, decision to publish, or preparation of the manuscript.

**Competing interests:** The authors have declared that no competing interests exist.

as maternity leave, cash payments and medical benefits, health protection at the workplace, employment protection (job security), non-discrimination, breastfeeding arrangements at work and coping with childcare [1]. In South Africa (SA), some elements of maternity protection are incorporated into national policy and legislation but the maternity protection policy landscape is fragmented and difficult to interpret [2].

The main stakeholders involved in ensuring that maternity protection at work is available in countries are governments (including departments of labour and social development), employers and employer organisations, and workers, usually represented by trade unions [3]. These three groups of stakeholders are referred to as the tripartite partners and together comprise the legislative and social security framework in countries. In addition, non-governmental organisations (NGOs) and civil society, including universities and research centres can be important allies in advocating for maternity protection.

Globally there are over 76 million domestic workers, representing between 1–2% of the global workforce [4]. Most domestic workers (76%) are women and around 80% of domestic workers globally work informally [4]. Although there have been attempts to formalise the domestic work sector, most domestic workers in SA are in positions of non-standard employment.

Characteristics unique to the domestic work employment relationship are that it exists within a private household and usually falls outside of conventional regulatory frameworks in many countries [5]. Domestic work is also considered vulnerable because incomes are usually low, and workers often do not have access to basic labour rights like employment contracts and employment benefits (such as pension contributions and paid leave). Recent research in SA has documented human rights violations against domestic workers including verbal abuse and harassment, denial of the rights to privacy and family life and discrimination on the grounds of pregnancy [6]. Historically, there were inadequate laws protecting the domestic work sector in SA. Since 1994, many laws and policies have been developed but these are not all appropriately implemented and there are still examples (such as legislative delays in access to social insurance and the national minimum wage) where the domestic work sector has been treated differently to other sectors. Domestic workers currently comprise 5.4% of the workforce in SA and the majority of domestic workers (97%) are women [7].

Some elements of maternity protection are available to certain categories of non-standard workers in SA. However, components of maternity protection are dispersed through many documents with weak government alignment on maternity protection and inadequate implementation, monitoring, and enforcement of existing maternity protection policy. Since domestic workers' place of work is a private household, this makes monitoring of labour law compliance especially challenging [8].

The focus of maternity protection research globally has been on the costing and affordability of paid maternity leave (i.e., maternity leave and cash payments while on maternity leave) and breastfeeding breaks [9–12]. Research has also focused on the provision of childcare, although this has not always been clearly considered in the context of maternity protection [13, 14]. Some research has been done on the awareness and perceptions of maternity protection by formally employed women in Vietnam [15], and employer perceptions of maternity leave and flexible working arrangements in the UK [16]. In SA, the National Department of Employment and Labour (NDEL) commissioned research into employees' knowledge of maternity rights [17] and other researchers have documented experiences of managers and mothers regarding workplace breastfeeding practices [18]. The Law Reform Commission documented gaps in maternity benefits for women in the informal economy in SA [19]. However, there has been no formal evaluation that we are aware of, globally or in

SA of the knowledge and perceptions of key stakeholders involved in ensuring that maternity protection is available and accessible. Stakeholders have some power and influence over availability of and accessibility to maternity protection. Therefore, this study sought to explore the knowledge, understanding and perceptions of key stakeholders in government, trade unions, non-governmental organisations (NGOs) and other relevant organisations of the maternity protection entitlements that should be available and accessible to female domestic workers in SA.

## Materials and methods

This was a qualitative, cross-sectional study that involved individual in-depth interviews (IDIs) with 15 participants. For this research, stakeholders working at either the national or sub-national (provincial) level in SA were selected. The sub-national stakeholders were selected from the Western Cape province, which is the third most populous province in SA and houses the legislative capital of SA.

### Study setting

South Africa is a middle-income country with high rates of poverty, inequality and unemployment [7]. In 2019, approximately 18 million South Africans (almost one-third of the population) were receiving some sort of social assistance from government [20]. At the beginning of 2022, 68.3% of working women were employed in the formal sector, 15.2% in the informal (non-agricultural) sector, 12.7% worked in private households and 3.8% in agriculture and of all working women, 12% were domestic workers.

### Ethics approval

All participants provided verbal informed consent for the individual IDIs and agreed to interviews being audio-recorded. Interview data was stored electronically and securely by CPK. Participants' confidentiality was maintained by removing personal information and names linked to individuals' insights from the transcribed data in the reporting of the results. Privacy, confidentiality, and anonymity was ensured. Ethical approval was obtained from the University of the Western Cape's Senate Research Committee and Ethics Committee [Reference Number: BM20/5/7].

### Participant sampling and selection

A stakeholder identification process was used to identify key individuals to take part in semi-structured individual IDIs. We started by consulting the ILO Maternity Protection Resource Package Module 4: 'Who are the main stakeholders?' [3] and reviewing policies that described any categories of stakeholders relevant to non-standard workers. Next, specific stakeholders from various sectors were identified using existing networks. Lastly, a snowball approach was used to identify additional potential participants. Stakeholders were purposively selected based on the knowledge they were likely to have on maternity benefits and their potential to influence the availability and accessibility of maternity protection benefits for female domestic workers. It was anticipated that stakeholders would include representatives from government (Department of Labour, Department of Health); trade unions, civil society organisations (such as NGOs that advocate for domestic workers labour rights); and companies (as they assist in accessing labour rights, such as claiming from the Unemployment Insurance Fund (UIF)). Stakeholder identification is an ongoing and iterative process and was refined throughout data collection [21]. After conducting interviews with 15 participants from various sectors, it was

felt that themes were starting to be repeated and therefore sufficient interviews had been conducted.

**Data collection.** A semi-structured interview guide was used to guide the IDIs (S1 Text: Interview Guide). Guiding questions were developed and adapted from sample interview guides available in the ILO Maternity Protection Resource Package [22]. The IDIs were conducted between October 2020 and July 2022. All interviews were conducted in English as all stakeholders were fluent in English. All except for one follow-up interview took place using an online/virtual platform of the participant's choice (Zoom or Microsoft Teams). When possible, the video function was used together with audio. Interviews ranged between 35 and 65 minutes but were on average 45 minutes long. Most interviews were conducted in one session but for two participants, follow-up interviews were conducted. All interviews were conducted by CPK, a South African female PhD student trained in qualitative research and then transcribed.

## Data analysis

The IDIs were analysed manually by one researcher (CPK). CPK coded the data and TD/MF read samples of the interview transcripts to confirm themes. A thematic analysis approach was used [23, 24]. Analysis began with familiarisation with the transcript contents (reading and re-reading of transcripts). Next, codes were allocated to similar groups of information. Then, initial themes were generated, which were then reviewed, developed, and refined. Finally, four main themes were decided on and the overarching themes were linked to the codes. A reflexivity journal was kept documenting any personal characteristics of the researcher that may have influenced the analysis process.

## Results

The sample characteristics are described in Table 1. Through the participant selection process, most participants (n = 13) were in positions of national reach and influence, while two operated at a sub-national (provincial) level.

Analysis of the interviews led to the identification of four major themes. The major themes and sub-themes are described in Fig 1.

**Table 1. Sample characteristics.**

|  | Total (N = 15) |
| --- | --- |
| Sex | |
| Female | 6 |
| Male | 9 |
| Sector | |
| Government | 5 |
| Trade union or professional association | 3 |
| Private company | 3 |
| Civil society | 2 |
| Independent labour organisation | 1 |
| UN Agency | 1 |
| Race/ethnicity | |
| Black African | 8 |
| Mixed race | 3 |
| White / Caucasian | 4 |

**Theme 1: Stakeholders have limited knowledge and understanding of comprehensive maternity protection**
- Maternity leave and cash payments were the maternity protection components most familiar to participants.
- Maternity protection rights are linked to other labour rights.

**Theme 2: Challenges in accessing cash payments while on maternity leave**
- Many general challenges related to accessing cash payments through the UIF while on maternity leave.
- Access to the UIF depends disproportionately on the employer.
- COVID-19 had a negative impact on service delivery of NDEL and therefore access to UIF.

**Theme 3: Labour-related characteristics unique to the domestic work sector**
- Working for an employer who is an individual can be problematic.
- It is difficult for unions to represent domestic workers.
- The monitoring and enforcement of labour laws in the domestic work sector has unique challenges.

**Theme 4: Suggestions for how cash payments during maternity leave could be improved**
- Increased awareness about national labour legislation and NDEL services.
- Improved functioning of the NDEL and UIF.

NDEL, National Department of Employment and Labour; UIF, Unemployment Insurance Fund

**Fig 1. Themes and sub-themes related to knowledge, understanding and perceptions of stakeholders in South Africa on accessibility and availability of maternity protection for domestic workers.**

## Theme 1: Stakeholders have limited knowledge and understanding of comprehensive maternity protection

Several stakeholders, such as those working for companies that assist women with their maternity claims from the national social insurance scheme, had good knowledge about most components of comprehensive maternity protection, while others had in-depth knowledge related to one or more components of maternity protection they were specifically involved in through their position or organisation. Almost all stakeholders described domestic workers and their employers generally having poor knowledge and understanding of maternity protection policy and legislation. It was described that maternity protection is complex and certain maternity protection rights are interconnected with other rights.

**Maternity leave and cash payments were the components of maternity protection most familiar to participants.** Almost all participants described **maternity leave** as the most common component of maternity protection available (especially since it is not mandatory for it to be paid in SA). There seemed to be confusion about whether maternity leave is three or four months, and several issues were raised related to maternity leave access. For example, domestic workers who cannot access cash payments while on maternity leave may not be able to afford to take advantage of their full maternity leave entitlements. One participant reported that some domestic workers are responsible for finding a temporary replacement to work for them while they are on maternity leave. It was also described that certain groups of domestic workers (e.g., migrant, or foreign workers) may be more vulnerable to not receiving a standard component of maternity protection, such as maternity leave.

Responses from participants clearly focused on one specific component of maternity protection, namely **cash payments** while on maternity leave, which is mainly provided by social insurance, through the UIF in SA. One participant described: *"Because when we think about, maternity, we often sometimes just think about the time and the payment." (S10, female, from a UN agency)* Almost all participants described how access to cash payments through the UIF while on maternity leave amongst domestic workers is limited. Many issues related to various

aspects of accessing cash payments while on maternity leave were described by participants and these are described in depth in the second theme. Several participants were unsure of the exact percentage of previous earnings that is paid out by social insurance when on maternity leave. One participant understood maternity protection to be cash payments during pregnancy and spoke extensively about the proposed 'maternal support grant being considered by the national Department of Social Development as an imperative of the National Development Plan.

Almost all participants described that **job insecurity** and **discrimination** due to pregnancy among domestic workers are high. Stakeholders described that many domestic workers fear losing their jobs if employers discover they are pregnant. One participant described how her own domestic worker hid her pregnancy from her and went on leave without informing the employer that it was due to childbirth: "*. . .because these things, even leave, I mean my, my helper went on leave, and she hid that she was pregnant from me.*" *(S10, female, from a UN agency)*. The same participant suggested that some domestic workers may resort to extreme measures such as not carrying the pregnancy to term (i.e., terminating pregnancy) to avoid risking losing their job: "*Some even don't carry the pregnancy to birth because they are afraid of losing their jobs*" *(S10, female, from a UN agency)*.

Another participant explained that some employers of domestic workers will directly state that they do not want the domestic worker to become pregnant:

"*I had some few domestic workers who called me that they were dismissed. I remember one said the employer said she doesn't want people pregnant. 'You're pregnant? Then you go, because I don't want people who fall pregnant here, working for me'.*"

(*S7, female, from a trade union*)

The same participant relayed that a domestic worker may fear losing her jobs if she asks to be registered with the UIF and/or the Compensation of Occupational Injuries and Diseases Act (COIDA):

"*Because. . . I am scared to approach my employer, register me with COIDA, because my employer will say, I don't have money to pay you and to register you. The gate is open. Your work is finished, because there are employers that, when domestic workers are approaching, they are like that, they don't want to work anymore telling me about the UIFs. So, you can go, I'm not going to register you with a UIF.*"

(*S7, female, from a trade union*)

COIDA is South African legalisation that aims to provide statutory insurance for workers, through employer contributions to a fund that then allows workers to claim compensation in the event of a work-related injury, illness, or fatality.

The health protection and medical care components of maternity protection were not described by participants. When directly asked about these benefits, it appears that opportunities for **medical care** (e.g., antenatal or postnatal check-ups) are not automatically guaranteed for domestic workers. Several participants, including the representatives from the domestic worker unions, described that access to medical care for domestic workers could be negotiated with employers and domestic work may allow some flexibility, where a domestic worker could go to the clinic and then come to work, starting later. Alternatives described were that annual leave could be used for antenatal check-ups, or that a domestic worker could attend the clinic on her off day (i.e., making use of unpaid leave).

When asked about maternity protection entitlements that should be available, none of the participants voluntarily mentioned **breastfeeding or expressing breaks**. Since breastfeeding breaks was part of the research question, the interviewer probed around this, and certain aspects of breastfeeding or expressing breaks were discussed by participants. Participants perceived that it was not common for domestic workers to either breastfeed their child while at work or take breaks to express breastmilk. One participant (working for a civil society organisation advocating for rights of various groups, including domestic workers) described that she would not have considered breastfeeding or expressing breaks as a right for any woman (not just domestic workers) returning to work after delivering a baby. After being questioned about this during the interview, this participant felt that awareness on the right to breastfeeding or expressing breaks is actually needed for all women, not just non-standard or domestic workers. Another participant (who worked for a company that assisted women with cash payments from the social insurance scheme) felt that breastfeeding or expressing breaks are not something usually considered for domestic workers: "*When you're breastfeeding. . . I mean, I'm moving a bit out of the domestic worker space, because I've never heard it talked about in the domestic worker space, but, you know, a place to breastfeed, obviously*" (S3, male, from a private company).

Participants also described that **childcare** is usually the responsibility of the domestic worker to organise and that while it is dependent on the individual employer, it is uncommon for domestic workers to bring their child to work. One participant described that often domestic workers leave their child with the maternal grandmother who may be far from where the domestic worker lives and/or works.

A participant from a civil society organisation campaigning for worker rights implied that comprehensive maternity protection may be an idealistic set of rights/entitlements and that certain rights (the right to fall pregnant and maintain job security, maternity leave) need to first be ensured before an entitlement such as breastfeeding or expressing breaks are considered. It also appeared that several participants (including two government sector stakeholders) perceived a hierarchy to maternity protection rights, where certain entitlements, such as maternity leave, were described as 'constitutional' or 'a core right' and 'cannot be varied' or are 'unlikely to be transgressed' and have stronger protection. In contrast, the right to breastfeeding or expressing breaks was interpreted to be more dependent on the relationship between the employer and worker and flexibility of the employer.

**Maternity protection rights are linked to other labour rights.**   Several participants, including a stakeholder working for a company that assists women with their maternity UIF claims, described how access to different labour-related rights (for example, unfair dismissal, maternity leave, and minimum wages) are intertwined. It was described that change in other legislation such as the national minimum wage bill can influence certain aspects of maternity protection. For example, if a domestic worker is paid below the national minimum wage, then her contributions to and claims from the UIF will be unacceptably low and it was questioned if this is currently monitored.

Two stakeholders working for domestic worker unions described how they felt that domestic workers have been historically excluded from labour policy in SA. These participants compared difficulty in accessing the UIF to inaccessibility to the national minimum wage and COIDA. The reason described for this is that domestic workers were only able to access the UIF after the passing of the Sectoral Determination for the domestic work sector and therefore domestic workers were structurally only able to access the UIF later than other workers. With the national minimum wage, domestic workers were initially at a lower minimum payment threshold compared to other workers and it was only after a few years that domestic workers had the same national minimum wage as others. In order for domestic workers to access COIDA, there needed to be a constitutional court case. This was described by one participant:

"*But, yes, we have suffered over the years, where domestic workers did not have maternity benefits, and a domestic worker did not really benefit from the payment, or all domestic workers was not registered. . .. this is how we see domestic workers are left out, in the cold. . . even now and the current national minimum wage, domestic workers is not getting the national minimum wage, we are excluded, from the national minimum wage. . . The same as we look at COIDA. We had to go to court, why did we have to go to court to prove that its unconstitutional? Now we're having this fight with the wages.*"

(S2, *female, from a trade union*)

### Theme 2: Challenges in accessibility to cash payments while on maternity leave

Access to cash payments while on maternity leave in SA is mainly facilitated through the national social insurance scheme (the UIF) managed by the NDEL. Participants listed and described many operational challenges of the UIF. Many of these are relevant to various categories of workers, but for certain reasons are more pronounced for domestic workers. The challenges will be described within the context of the domestic work sector.

**Many general challenges related to accessing cash payments through the UIF while on maternity leave.** Almost all participants reported that the UIF is not a user-friendly system, describing that it is complicated and time-consuming. Participants described that the NDEL and UIF are inefficient with regular disruptions in service and bottlenecks at various stages in the claiming process. The word 'mess' was used to describe the UIF, with one participant expressing frustration and exclaiming: "*It's a mess, because the UIF is just, that's what it is, they're a mess!*" (S3, *male, from a private company*). Another participant summarised that "*Currently, I think, there's a mess-up.*" (S2, *female, from a trade union*). While it seems that the main steps required for registration and contribution to the UIF can all be done physically at Labour Centres or online via the online system (uFiling), several participants had negative experiences with the online system, with one participant describing making payments that were not reflected. Others felt that the online system (uFiling) was good but remains inaccessible to certain workers (such as many domestic workers) without internet or a smartphone. Almost all participants described that most domestic workers are not registered with the UIF for multiple reasons including that it is not easy to register with the UIF. Again, certain subgroups such as migrant workers (foreign nationals) were described as having additional obstacles (for example needing to submit additional documentation) when registering for the UIF.

One stakeholder described that there are inconsistencies in the way certain policies are implemented and that terminology in policy documents may be interpreted differently:

"*So, there's massive variations of policies within the UIF, the main thing is terminology that they use, which is not very understandable to the lay person, or to the person that is on, the man on the street. We've had to familiarise ourselves with that. There's also underwritten laws within UIF, that they don't let the public know. . . so they'll refuse your claim based on that sub-section. And that's something that's really frustrating to the public because they're not aware of the sub-ruling.*"

(S1, *male, from a private company*)

An example of this, described by two participants is that some women have been told that they should be able to start the UIF claim process when they are pregnant so that the payment can be received as soon as the woman goes on maternity leave. In practice, women are only

able to submit their claims once the baby has been born which often results in delayed payments.

> *Respondent: There's again, varying, I'm not going to say laws, but varying, articles around this. So, the UIF generally say, and they will say, you can submit a claim 6 weeks prior to the start of maternity leave. Um, this isn't true. Ok. You can only submit your application from the first date that you are not receiving a hundred percent salary. The reason for that,*
>
> *Moderator: So that could be before the child is born?*
>
> *Respondent: Yes, so you can submit before the birth of your child, but you have to submit only from when you are not receiving a full salary.*
>
> *Moderator: Ok. And sorry, you were going to say, the reason for that,*
>
> *Respondent: Ja, the reason for that, is that, you're claiming from the unemployment fund. And, by claiming from the unemployment fund, technically, you have to be unemployed, or not getting paid in full.*
>
> *(S1, male, from a private company)*

Participants described that many women must go back to 'Labour Centres' (local NDEL offices) repeatedly to follow up on applications or submit additional forms they might not have known about. When at the labour centres, stakeholders described that many women wait in queues for hours, often needing to take an entire day to submit their claims for cash payment while on maternity leave. Because the claims for cash payment can only be submitted after childbirth, some women need to go back to work while they are waiting to receive their cash payments. Women that have returned to work sometimes have to take unpaid days off work to submit their claims, meaning that women are losing income in their attempt to access cash payments while on maternity leave. As one stakeholder described: *"...there's people who phone me crying, they say they've been to the labour centre eight times, and every time they've been sent away and they just don't know what to do anymore..."* (S3, *male, from a private company*).

It seems that there are more forms required for maternity claims (such as a form signed by a medical practitioner confirming a woman's delivery date) than for other unemployment claims. Various stakeholders described that to claim 'maternity benefits' from the UIF, many forms need to be completed by different people:

> *"There's just so much, I remember when my wife did it, we actually did it together, we sat at the table, we had all the forms spread out on the table, it's a mess, I mean, you don't know what's left or, where left and right, is, and we did, all the mistakes [laughs] which, you know, I was rolling my eyes now, and it's just so complex to complete those documents. For anyone, and the employer has to do, 3 forms, and many domestic employers just have absolutely no idea how to do that, I mean, many big corporates have no idea how to do that."*
>
> *(S3, male, from a private company)*

Participants described that it could take months for UIF registrations to be reflected. Almost all participants reported that the result of these inefficiencies, disruptions and bottlenecks is that often, claims for payments while on maternity leave are delayed. A government official described being embarrassed when he hears about women who have waited a year to receive a maternity claim:

*"There is a long struggle, I mean you can wait up to a year. Especially on maternity, you know how embarrassed I am when somebody comes, with a child. To enquire about the money, and they have not received a cent from the state yet."*

*(S8, male, provincial government official)*

One participant (a representative from a domestic worker trade union) described how many women go into debt (making use of microlenders) while on maternity leave and waiting for the UIF payments to cover basic household expenses:

*"You see, that is where, this, financial lenders, this people that's lending money, this people that's making it so easy for domestic workers, to just pick up the phone, and you can borrow. So, what happens? The domestic worker goes and borrows money from a financial institution, or this money lenders. At the end of the month, when she gets that payment, she needs to pay nearly, you know, half of that, back to them, which means now, she borrows again. She borrows again. So, at the end of that financial four months, she finds herself deeply in debt to a financial institution, she needs to go work for her full wages, but half of it go back."*

*(S2, female, from a trade union)*

This means that these women will now need to pay money back to lenders with added interest, increasing their monthly expenses which places already vulnerable women in an even more precarious situation.

This same participant also described how the current cash payments are insufficient for domestic workers:

*"I think it is something also, we have to look at that clauses in this unemployment fund, and see, what is really benefitting the domestic worker, and what is not. . . And I think this is the reality of the domestic workers is that it's there, yes, but how accessible is it to them? And can they survive on the payment that they get?"*

(S2, *female, from a trade union*)

While most of the comments made about the NDEL and UIF were negative, one stakeholder displayed empathy for NDEL officials: *"A lot of them are trying their best, and there's a lot of good apples within, the Department of Labour, that work. . . late hours, and we see emails from individuals at 11'o'clock at night".* (S1, *male, from a private company*) Therefore, although most information provided about the functioning of the NDEL and UIF was negative and critical, there was some positive feedback.

**Access to the UIF is disproportionately dependent on the employer.**   Access to the UIF (and therefore cash payments while on maternity leave) is conditionally dependent on the employer having registered the worker or employee, the employer submitting employment declarations, and the employer having contributed to the UIF. All three steps are reliant on certain actions by the employer. This is a substantial barrier to access for many domestic workers. Not all domestic worker employers are familiar with relevant legislation or understand that domestic workers are eligible to register for UIF. The result of this is that many domestic workers cannot access cash benefits while on maternity leave, as described by one participant: *"The problem at the moment is that you as an employee, when you go to the UIF, you are being penalised, by not getting your claim approved, you're not getting your money, because your employer messed up." (S3, male, from a private company)* This therefore means that there is dual responsibility in making this component of maternity protection available to domestic

workers, whereby access depends on both a functioning government department (the UIF/NDEL) and employers who understand their obligations.

**Impact of COVID-19 on service delivery by NDEL (and therefore access to UIF).**
Almost all participants described how the COVID-19 pandemic disrupted services at NDEL and shifted priorities, specifically resulting in increased processing time of claims, increased time to respond to queries, and a disruption in face-to-face services (including users not being able to walk into offices and department officials not going into communities to do advocacy). The result of the physical distancing required by the pandemic was a shift to the virtual platform already available, and while this did work for some, not all domestic workers can access online technology. A few participants felt that the COVID-19 pandemic simply highlighted existing inefficiencies within the NDEL. Others felt that the pandemic presented some opportunities, such as the NDEL improving the online systems and online access and realising that certain manual processes could be removed, and efficiency of services improved.

## Theme 3: Labour-related characteristics unique to the domestic work sector

When describing availability of and accessibility to maternity protection for domestic workers, almost all stakeholders described that there are certain unique characteristics of domestic work related to labour legislation (i.e., context) that needs to be considered because it makes access to certain components of maternity protection challenging.

**Working for an employer that is an individual has challenges.** While some domestic workers work full-time for one employer, many domestic workers work for multiple employers sometimes working for different employers on different days of the week. One stakeholder described that the nature of domestic work in SA has changed whereby in the past, many domestic workers worked for one household full-time and often lived on the property, but now not everyone can afford that, and some people can only have domestic help a few days a week on a part-time basis, without the domestic worker living on the property. The use of a "platform" (in the form of an application that can be accessed online through a smartphone or computer, also referred to as "the gig economy") to access domestic work services is also relatively new. Domestic workers can register themselves and employers can request domestic work services through this online platform on an *ad hoc* basis.

Working for a single employer could have potential advantages. It may be easier for one employer with one employee to ensure that all relevant labour related registrations are completed. Stakeholders reported that some employers do want to 'do the right thing' and comply with the rules (e.g., to register the domestic worker with the UIF). However, working for an individual employer can also be a disadvantage as that individual can have disproportionate power in determining whether the domestic worker can access certain labour/employment benefits/rights. Particularly between domestic workers and their employers, there exists uneven power relationships, as a few stakeholders described:

> *"They [domestic workers] are in positions of severe vulnerability, power disparity and its very uncomfortable for them, not to mention language issues, etcetera."*
>
> *(S3, male, from a private company)*

> *"Because there is just so much power that the employers have, and so much non-compliance when it comes to UIF that domestic workers don't even have a choice but to go back to work, like, as soon as possible, essentially".*
>
> *(S13, female, from civil society)*

Stakeholders described this power imbalance further by explaining that it is difficult for a domestic worker to challenge their employer if the employer has not registered or contributed to the UIF. Furthermore, individual domestic employers often do not consider themselves to be employers, which can result in many informal arrangements. These informal arrangements could be positive and result in flexibility and situations where, for example, a domestic worker may be able to bring a new-born baby to work with her. But other employers may not be as accommodating, creating a sector with diverse and heterogeneous employment relationships.

Having an employer that is an individual also means that the employer has limited financial capacity to be able to afford to provide a full salary and pay a temporary replacement worker while a worker is on maternity leave, as was described by one stakeholder:

*"So, the employer will be paying the domestic worker, who's working for me while I'm on maternity leave, and it is irrational, and I cannot expect the employer to pay the domestic worker that replaced me and also to pay me a salary while I'm on maternity leave."*

(S7, *female, from a trade union*)

One stakeholder described a major difference between domestic and commercial employers being that commercial employers have human resources specialists who are familiar with labour laws and deal with issues such as UIF registrations regularly.

*"The advantage that commercial employers will have over domestic is because for commercial employers there will be people that will be employed to take care of UIF related information. Maybe that submission is going to be happening on a periodic basis, even on a monthly basis for that matter, only to find out that in the domestic space, because the domestic employer might not be having the necessary time for them to be able to register and be able to submit the information of that employee. And because maybe the employer does not even have the full information around how to submit information through to the UIF, that means even at the time when the domestic employee falls pregnant, because the information will not be sitting on the UIF, they will not be in a position to actually declare the fact that my employee will not be working for the next few months because they're on maternity leave."*

(S11, *male, national government official*)

**Monitoring and enforcement of labour laws in the domestic sector is especially challenging.** While many challenges related to monitoring and enforcement of labour legislation in general were described, participants described that because a domestic worker's place of work is a private home, this makes monitoring and enforcement especially difficult. The main route of monitoring labour laws in SA is by labour inspectors visiting places of work and reviewing documents like written employment contracts, proof of contribution to the social insurance scheme (the UIF) and observing other labour-related practices. Participants described various reasons why doing this monitoring in a workplace that is a private home is especially challenging. Firstly, it is difficult to target households to monitor because not all households employ a domestic worker. Therefore, random selection inspections are problematic, because inspectors would not know which households to target. Secondly, a private home-owner has the right to deny a person entry into their house. Thirdly, employers of domestic workers are often workers or employees themselves, so might be at their own workplace during working hours and therefore might not be present at the domestic workers workplace at the time when a labour inspector would usually visit. This makes it difficult for inspectors

(enforcement officers) to do an inspection or to communicate with domestic worker employers. One stakeholder described that to circumvent this, labour inspectors would need to do inspections for domestic workers after hours (e.g., on a weekend or public holiday) but the NDEL does not have the structures in place to do this:

> *"The first challenge that we will have, which is a challenge with access, meaning that for us to be able to get access to the domestic employer, we need to do our inspections during the weekends and during public holidays because the domestic employer is not working, and they are at home. And because we are in the form of the employment, which is in the public sector stage, during the weekend, I am also at home resting."*

> (S11, *male, national government official*)

Strategies suggested to increase employer responsibility included offering incentives to employers to comply with legislation, for example offering rebates on UIF contributions. An alternative suggestion to hold employers accountable was to enforce penalties for non-compliance, such as fines for those who do not register their domestic worker for the UIF. One participant compared compliance with the UIF Act to other government programmes by explaining:

> *"I think it can be done, we pay taxes, we pay our. . . tolls. But there are ways to make these things, there's a creative solution to all of this. I just think we need to sort of apply our minds to it."*

> (S13, *female, from civil society*).

While one stakeholder felt that increasing the number of employees registered with the UIF would increase the UIF's income and therefore make more funds available to assist the NDEL with fulfilling their obligations (S8), another stakeholder believed the UIF has a surplus of funds and that the problem is rather the misuse of funds (S1). Several stakeholders agreed that increasing the numbers of and building capacity of labour inspectors could improve the functioning of the social insurance scheme.

One participant described that currently, the only time that government really gets involved in implementing penalties is through the Commission for Conciliation, Mediation and Arbitration (CCMA) once there is a dispute or if there has been an unfair dismissal. It was stated that remedial actions are needed at various stages in the process and not only once the point of dismissal has been reached.

### Theme 4: Suggestions for how cash payments during maternity leave could be improved

Almost all participants that provided suggestions for how access to maternity protection could be improved made suggestions to improve the functioning of the UIF and therefore better access to cash payments from the social insurance scheme while on maternity leave.

**Increased awareness about national labour legislation and NDEL services.** Participants agreed that advocacy is needed to provide education and share information on domestic worker rights, employer responsibilities, current legislation, the necessity of registering with the UIF and the availability of online services. Participants indicated that the target for this information provision should be policy makers, NDEL staff, employers, and workers, and that the NDEL should make a concerted effort to reach different categories of workers. Many suggestions for awareness raising were provided, including roadshows, issuing pamphlets at

community settings (e.g., churches) and to employers, publishing advocacy papers, policy briefs and practical guidelines about maternity protection for domestic workers and allocating budget to a media drive advertising NDEL services. Several participants from various sectors suggested that when information on maternity protection rights is shared, that this could be linked to other similar legislation (e.g., the national minimum wage bill, Occupational Health and Safety (OHS) and COIDA). There were specific suggestions made for information targeting employers, including that domestic worker employers need to realise that they are employers, even if they are only employing one person in their household, and should know and fulfil certain obligations of being an employer, such as providing the domestic worker with a contract and registering the domestic worker with the UIF and COIDA. It was suggested that information about the obligations of domestic worker employers could be shared widely by different media platforms using a very direct approach:

> "*That could be a really great first step is if that is a priority, whether it's community, radio stations, national radio stations, billboards, television, all sorts of different media to communicate to employers that they have certain obligations in respect to their employment of a domestic worker in their home. Billboards saying please register your employees for UIF, with COIDA, a little bit more sort of explicit. Like if you employ a domestic worker, you need to register them with the compensation fund.*"

> (*S13, female, from civil society*)

Several participants, including a representative from a domestic worker's union, emphasised that employers should find out what their responsibilities are such as registering their domestic worker with the UIF and COIDA. As one participant described *"it can't just be everything is relied on poor working women to kind of sort themselves out"* (*S9, female, from civil society*).

**Improved functioning of the NDEL and UIF.**   Almost all participants described that the NDEL and UIF systems need to be strengthened, especially that it should be easier and faster for employers to register workers with the UIF. One participant described that sometimes domestic worker employers need to take leave from work to go to NDEL offices in person to register a domestic worker for the UIF. This is not feasible or sustainable. Examples provided by participants to improve this included being able to complete the UIF registration process on a mobile phone and having a "one-stop-shop" system where employers can register for the UIF, COIDA and any other declarations all at once. This would involve developing a way to link all departments within the NDEL so that information could be shared between departments and not need to be resubmitted individually.

Participants described that employers should not have all the control as to whether a worker is registered with the UIF. One participant described that *"you have to disrupt this power dynamic of employers having all the control as to whether a registration is made or not"* (*S9, female, from civil society*). A suggestion to resolve this issue was to enable worker-led registration with the UIF but participants also cautioned that any change in registration should not shift the burden of compliance to the worker. The participant suggested that *"Workers should be allowed to register as a domestic worker and then that should mean that immediately there's a trigger created on the employer, who is then either incentivised, through carrots or sticks, to contribute to that, system."* (*S9, female, from civil society*). It was, however, acknowledged that such a change would require legislative reform. Other solutions that would probably also require legislative change included making it compulsory for employers to register with an employer association who could then facilitate registration.

Other processes suggested by participants related to the organisation of labour in the domestic work sector. One participant suggested the setting up of a bargaining council specific for domestic workers. Almost all participants described that improved unionisation of the domestic work sector could improve accessibility to maternity protection. Reasons provided for this were that unions could provide a channel for communication and education between domestic workers and their employers and that domestic worker unions could collaborate with bigger unions and receive support from those bigger organisations.

## Discussion

This research aimed to explore the knowledge, understanding and perceptions of stakeholders in various sectors, of the maternity protection that should be available and accessible to domestic workers in SA. It appears that most stakeholders' knowledge and understanding of maternity protection are limited to the component/s of maternity protection they are involved in ensuring availability and/or access to, with maternity leave and cash payments being the most familiar. Significant challenges in domestic workers' access to cash payments while on maternity leave were described including general challenges with the national social insurance scheme, dependence on the employer for access to social insurance and compounded negative consequences of the COVID-19 pandemic. There are labour related characteristics unique to the domestic work sector that impede access to maternity protection, including the employer usually being an individual and monitoring and enforcement of work conditions that take place in private households being challenging. Participants made many suggestions for how access to maternity protection for domestic workers could be improved, including increasing awareness around maternity protection entitlements and improving the functioning of the NDEL and UIF.

Stakeholders interviewed in this research were most familiar with the maternity leave and cash payment elements of maternity protection. These components have also been the focus of previous research conducted in other countries on maternity protection [25–27]. Results however show that stakeholders are not familiar with all components of comprehensive maternity protection in SA. Notably, stakeholders are unfamiliar with breastfeeding breaks and opportunities to access childcare as provisions of maternity protection. Also, they did not discuss health protection at the workplace or medical benefits while on maternity leave, nor non-discrimination and employment protection (job security). This is a problem, because all components of maternity protection need to be made available and accessible to working women for the full benefits of maternity protection to be realised. To ensure that comprehensive maternity protection is available and accessible to non-standard workers such as domestic workers, improved knowledge and understanding of all components of comprehensive maternity protection by the key stakeholders assigned responsibility for maternity protection is required. The provision of breaks for breastfeeding or expressing breastmilk for domestic workers should be something that could be quite feasible and easy to implement for domestic workers, since their place of work is a private household and is therefore conducive to either bringing the baby with the mother to work or having a small private space available for breastfeeding or expressing. There should not be additional expense or infrastructure required for making breastfeeding or expressing breaks available to domestic workers, and the flexibility of domestic work should enable the provision of such breaks. For future maternity protection campaigning and advocacy, it is important to refer to comprehensive maternity protection and not selectively promote individual components of maternity protection. Furthermore, it may be helpful in practice to link access to maternity protection rights to the access to other rights such as the national minimum wage and the compensation for occupations and injury on duty

as was raised by several participants. There have been some efforts to do this in SA, through the organisation of dialogues and roundtable discussions bringing together relevant stakeholders to discuss issues like the unemployment insurance fund and the compensation fund together [28]. The domestic work sector globally it still relatively unorganised and stronger unionisation of the domestic work sector could also result in better advocacy for maternity protection.

Stakeholders described many challenges that domestic workers experience in accessing cash payments while on maternity leave and these highlight inefficiencies in the functioning of the UIF in SA. Challenges in accessing social protection for domestic workers have been reported in other African countries such as Nigeria [29] and Zambia [30]. Domestic workers globally face multiple barriers to legal coverage and effective access to social security, especially in Africa, Asia and the Pacific and the Arab States, which are regions employing large numbers of domestic workers [31, 32]. A recent ILO report on policy trends, statistics, and extension strategies for access to social security for domestic workers described how burdensome administrative procedures can reduce social security coverage by increasing the transaction costs for employers and domestic workers in accessing social security [32]. These transaction costs were described as the time and resources spent complying with administrative requirements, which is similar to what was described by stakeholders in this study. It therefore appears to be important that the administrative barriers to accessing social insurance in SA need to be removed, and the dependence on employers for domestic workers to access cash payments needs to be reduced. The ILO has also recommended that legal reforms together with improved governance is important to see improvements in access to social security for domestic workers [32]. Simplified and affordable contribution mechanisms for employers have been previously suggested in another African country, Zambia [30]. Since workers need to be working a minimum of 24 hours a month for an employer to contribute to the national social insurance scheme [33], it would also be important for there to be safety nets available to those working less than this threshold.

The current cash payment provided by the UIF is 66% of previous earnings. For women earning already low incomes—like domestic workers, some of whom do not even earn the national minimum wage—this is a percentage of an already low wage and is insufficient at a time when expenses increase due to increased household size. While the ILO Maternity Protection Convention states that cash benefits while on maternity leave should not be less than two-thirds of a woman's previous earnings, the same convention also states that cash benefits should ensure that women can maintain themselves and with a suitable standard of living. Furthermore, the convention is a minimum requirement, and the ILO Maternity Protection Recommendation 191 goes further to state that cash benefits while on maternity leave should be raised to the full amount of women's previous earnings. Participants in this study described how the already inefficient social insurance service by government was unfortunately made worse by the COVID-19 pandemic. For domestic workers, this amplification of existing challenges due to the COVID-19 pandemic took place at a global level, where social protection coverage gaps experienced by domestic workers were made even more apparent due to the pandemic [32, 34].

Often a domestic worker is a single employee working for a single employer, in the employers' private household space. A consequence of being a single employee at a workplace is that actions taken by other sectors such as protesting or negotiating through collective bargaining are not effective for domestic workers [35]. The domestic work sector is also difficult for unions to represent. Working in an employer's private household can increase the intimacy of the working relationship and some domestic workers even live at their place of work. All these characteristics contribute to a complex power dynamic, where access to labour rights including

maternity protection is dependent on the nature of quite a personal relationship between the employer and employee and make it very difficult to monitor and enforce labour law in the domestic sector. Other characteristics unique to domestic work are that most domestic workers are women and many are of reproductive age, struggling to balance their own family responsibilities while often looking after other people's children as part of their jobs and unable to access maternity protection [36].

Sometimes domestic workers have limited access to their own children due to working far from where they live, and many domestic workers are migrant workers who may not be documented which means they cannot register for national schemes like the UIF. India provides an example of the protection of migrant workers through social security agreements that have been developed with eight other countries to provide social protection to Indian workers employed in those countries to ensure equality of treatment and avoidance of duplicate coverage [35]. South Africa needs to do more to protect the many migrant workers, especially due to recent xenophobic attacks [37, 38] and this improvement should include access to social protection. The ILO has recently acknowledged some of the labour-related characteristics unique to the domestic work sector that make access to social security (including maternity protection) challenging, such as administrative barriers, lack of enforcement and low compliance and lack of information and organization amongst the sector [39]. Many possible solutions are provided by the ILO including removing administrative barriers, facilitating payment contributions, promoting compliance (financial and criminal penalties), raising awareness and disseminating information, building on social dialogue and collective bargaining, developing and implementing integrated and coherent policies and protecting migrant domestic workers [39]. Others also suggest that non-contributory schemes could assist in extending social protection to vulnerable groups like domestic workers and that social assistance may be required for low-income women [31]. While SA does provide social assistance, it may not be sufficient to ensure income replacement while on maternity leave. A more comprehensive approach to extending social protection including comprehensive maternity protection to domestic workers is required to ensure economic and social development for women.

The ILO has described how social protection has the potential to enable the transition of domestic worker from informality to formality [32]. Attempts have been made to formalise domestic work internationally and in SA. Globally, the ILO Recommendation R204 on the Transition from the Informal to the Formal Economy calls particular attention to vulnerable groups including domestic workers when strategies to transition for the formal economy are developed [40]. In SA, it appears that insufficient consideration has been given to this, as evidenced by the existence of a Sectoral Determination for Domestic Work for the past 20 years, yet still many domestic workers struggle to access the provisions included in this sectoral determination. A reason for this could be that the complex employment relationships between domestic workers and their employers often makes it difficult to determine where employer obligations reside [40]. The ILO has provided recommendations of how domestic work can be formalised and advised that incentives of formal arrangements should outweigh any benefits of informal employment [30]. Other countries have used a combination of deterrent approaches (including labour inspection, complaints mechanisms, dispute settlement systems together with advisory and support services) and enabling approaches (such as removing barriers to formalising work, ensuring worker and employer awareness, increasing the benefits of formal sector work via income tax deductions or tax credits, VAT reductions wage subsidies, lower social security contributions, etc.) and also simplification of procedures [30]. SA and other countries with high rates of informality could be encouraged to implement some of these approaches.

While the results from this research are quite particular to SA, the insights obtained could be relevant for maternity protection for domestic workers in other low- and middle-income countries. Only 6% of domestic workers globally have access to comprehensive social protection and half of domestic workers have no social protection coverage at all [32]. Furthermore, effective, or actual coverage is even lower than legal coverage since only 20% of domestic workers are covered in practice because most are employed informally [32]. There are many challenges to promoting decent work for domestic workers, including the heterogeneity (of both domestic workers and their employers) and distinctive features of the sector [41]. Unlike the Maternity Protection Convention C183, SA has ratified the ILO's Domestic Worker Convention C189 which states that "domestic workers enjoy conditions that are not less favourable than those applicable to workers generally in respect of social security protection, including with respect to maternity" [42, 43]. Therefore, considering this global commitment, stakeholders in SA need to ensure that equal working conditions for domestic workers becomes a reality, which may involve further efforts to ensure genuine formalisation of the sector.

## Limitations

While care was taken to identify and select a diverse group of stakeholders, there may have been some key stakeholders (e.g., a representative from the ILO) that were not included in the sample. The researchers attempted to reduce bias by ensuring reflexivity, but qualitative analysis still has the risk of subjectivity in the interpretation of the results.

## Conclusions

Women working in positions of non-standard employment, including domestic workers, are especially vulnerable to not being able to access comprehensive maternity protection. There are distinctive features of the domestic worker sector that make access to maternity protection challenging, especially access to cash payments while on maternity leave. Ensuring greater awareness of all components of maternity protection and improving implementation of existing labour legislation intended to guarantee maternity protection for non-standard workers in SA is important to improve access to social protections such as maternity protection for this vulnerable group. For equal availability of maternity protection for all working women may require formalisation of those employed in non-standard relationships. Improved access to maternity protection would contribute to optimal maternal and new-born health and contribute to economic security for women around the time of childbirth.

## Supporting information

**S1 Text. Interview guide for individual semi-structured interviews with key stakeholders.** (DOCX)

## Author Contributions

**Conceptualization:** Catherine Pereira-Kotze, Mieke Faber, Tanya Doherty.

**Data curation:** Catherine Pereira-Kotze.

**Formal analysis:** Catherine Pereira-Kotze.

**Funding acquisition:** Catherine Pereira-Kotze.

**Investigation:** Catherine Pereira-Kotze.

**Methodology:** Catherine Pereira-Kotze, Mieke Faber, Tanya Doherty.

**Project administration:** Catherine Pereira-Kotze.

**Resources:** Catherine Pereira-Kotze.

**Supervision:** Mieke Faber, Tanya Doherty.

**Writing – original draft:** Catherine Pereira-Kotze.

**Writing – review & editing:** Catherine Pereira-Kotze, Mieke Faber, Tanya Doherty.

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
