## [Decision Letter · Decision Letter 0]

6 Mar 2023

PGPH-D-22-01467

Knowledge, understanding and perceptions of key stakeholders on the maternity protection available and accessible to female domestic workers in South Africa

Dear Dr. Pereira-Kotze,

Thank you for submitting your manuscript to PLOS Global Public Health. After careful consideration, we feel that it has merit but does not fully meet PLOS Global Public Health’s publication criteria as it currently stands. Therefore, we invite you to submit a revised version of the manuscript that addresses the points raised during the review process.

It is important to address the suggestions made by the reviewers, particularly from Mellissa Withers, it will be more details of the interviews, these suggestions are minor and in general both reviewers agreed that it has sounded methodology and contributes to the literature in the area. 

We look forward to receiving your revised manuscript.

Kind regards,

María De Jesús Medina Arellano, PhD

Academic Editor

Journal Requirements:

2. Please provide separate figure files in .tif or .eps format only and remove any figures embedded in your manuscript file. Please also ensure that all files are under our size limit of 10MB.

4. In the online submission form, you indicated that "The datasets

generated and/or analysed during the current study are not publicly available

to protect participant confidentiality but are available in an anonymised form

from the corresponding author on reasonable request". All PLOS journals now require all data underlying the findings described in their manuscript to be freely available to other researchers, either 1. In a public repository, 2. Within the manuscript itself, or 3. Uploaded as supplementary information.

Additional Editor Comments (if provided):

Reviewers' comments:

Reviewer's Responses to Questions

**Comments to the Author**

1. Does this manuscript meet PLOS Global Public Health’s publication criteria? Is the manuscript technically sound, and do the data support the conclusions? The manuscript must describe methodologically and ethically rigorous research with conclusions that are appropriately drawn based on the data presented.

Reviewer #1: Yes

Reviewer #2: Yes

2. Has the statistical analysis been performed appropriately and rigorously?

Reviewer #1: N/A

Reviewer #2: N/A

3. Have the authors made all data underlying the findings in their manuscript fully available (please refer to the Data Availability Statement at the start of the manuscript PDF file)?

Reviewer #1: Yes

Reviewer #2: Yes

4. Is the manuscript presented in an intelligible fashion and written in standard English?

Reviewer #1: Yes

Reviewer #2: Yes

5. Review Comments to the Author

Reviewer #1: Thank you for the opportunity to review this manuscript. I was really intrigued with this study and found it very interesting. The paper is well-written and easy to follow.

The authors provide adequate background on the issue of domestic workers and their unique vulnerabilities in terms of health and well-being. I also appreciated the background on efforts to improve maternity protection globally and in SA. This contextual background helped me to better understand the results, and were also useful in justifying the importance of this study.

I think more insight into domestic workers’ health status makes an important contribution to the literature.

The methods section included mention of consent, and IRB approval. The authors explained the methods used for data protection. I also thought it was good to mention that the interviews were audio-recorded.

Interviews were conducted with key stakeholders until saturation point was reached. I think a snowball approach with purposive sampling made sense for this type of study.

Since the interviews were done with stakeholders (who I presume were educated), I wonder why only verbal consent was required. Also, I would like to know how the interview guide was developed and how many questions were included. What were the domains or types of questions asked?

I would also like to know more about the person who was the interviewer. What was this person’s training in qualitative research? Is this person a citizen of SA? I would also like to know if anyone else helped with coding. If not, that would seem like a major limitation of this study because it would be based on one person’s interpretation of the data.

Results: it would be helpful to know how many when the authors say words like “some stakeholders” or “many stakeholders.”

It would also be helpful to know more about the stakeholders when citing examples or direct quotes. For example, is this person male or female, from the national or provincial pool, etc., instead of just using terms like (S10).

I thought the results had some important implications in terms of providing better protections to pregnant migrant workers. Obviously there was a lot of information about benefits that was not being disseminated. If these stakeholders were not familiar with some of the protective policies, it is even less likely that workers will know about them.

I thought the authors did a good job of linking the results to other previous studies. I also liked the recommendations to improve the maternal health of domestic workers.

Reviewer #2: The manuscript explores the knowledge, understanding and perceptions of key stakeholders in government, trade unions, non-governmental organisations and other relevant organisations of the maternity protection entitlements that should be available and accessible to female domestic workers in South Africa. It is qualitative cross-sectional study that involved individual in-depth interviews with fifteen participants. In an accessible and methodical way, it contains the obstacles that domestic workers face for the recognition and protection of reproductive rights, specifically related to pregnancy, childbirth and postpartum. While rights such as maternity leave and cash payments were the components best known by the participants, there is little mention of important aspects (e.g. breaks for breastfeeding) . It requires certainty in specific rights, like the right to get pregnant and maintain job security. It also highlights the challenge of officially registering workers, the areas of opportunity when employers are individuals or companies. In the same way, the authors address the importance of strengthening the knowledge of maternity protection. There is a dual responsibility (employers and government) in making this component of maternity protection available to domestic workers.

6. PLOS authors have the option to publish the peer review history of their article (what does this mean?). If published, this will include your full peer review and any attached files.

**Do you want your identity to be public for this peer review?** For information about this choice, including consent withdrawal, please see our Privacy Policy.

Reviewer #1: **Yes: **Mellissa Withers

Reviewer #2: No

---

## [Editor Report · Decision Letter 1]

25 May 2023

Knowledge, understanding and perceptions of key stakeholders on the maternity protection available and accessible to female domestic workers in South Africa

PGPH-D-22-01467R1

Dear Dr Pereira-Kotze,

We are pleased to inform you that your manuscript 'Knowledge, understanding and perceptions of key stakeholders on the maternity protection available and accessible to female domestic workers in South Africa' has been provisionally accepted for publication in PLOS Global Public Health.

Best regards,

María De Jesús Medina Arellano, PhD

Academic Editor